# Therapeutics That Can Potentially Replicate or Augment the Anti-Aging Effects of Physical Exercise

**DOI:** 10.3390/ijms23179957

**Published:** 2022-09-01

**Authors:** Adriana De Sousa Lages, Valentim Lopes, João Horta, João Espregueira-Mendes, Renato Andrade, Alexandre Rebelo-Marques

**Affiliations:** 1Endocrinology Department, Hospital de Braga, 4710 Braga, Portugal; 2Faculty of Medicine, University of Coimbra, 3000 Coimbra, Portugal; 3Clínica Espregueira, FIFA Excellence Centre of Medicine, 4350 Porto, Portugal

**Keywords:** aging, anti-aging, longevity, drug, supplement, therapeutics, AMPK, mTOR, health span, senescence

## Abstract

Globally, better health care access and social conditions ensured a significant increase in the life expectancy of the population. There is, however, a clear increase in the incidence of age-related diseases which, besides affecting the social and economic sustainability of countries and regions around the globe, leads to a decrease in the individual’s quality of life. There is an urgent need for interventions that can reverse, or at least prevent and delay, the age-associated pathological deterioration. Within this line, this narrative review aims to assess updated evidence that explores the potential therapeutic targets that can mimic or complement the recognized anti-aging effects of physical exercise. We considered pertinent to review the anti-aging effects of the following drugs and supplements: Rapamycin and Rapamycin analogues (Rapalogs); Metformin; 2-deoxy-D-glucose; Somatostatin analogues; Pegvisomant; Trametinib; Spermidine; Fisetin; Quercetin; Navitoclax; TA-65; Resveratrol; Melatonin; Curcumin; Rhodiola rosea and Caffeine. The current scientific evidence on the anti-aging effect of these drugs and supplements is still scarce and no recommendation of their generalized use can be made at this stage. Further studies are warranted to determine which therapies display a geroprotective effect and are capable of emulating the benefits of physical exercise.

## 1. Introduction

Prior to the COVID-19 pandemic, we assisted to an improvement on global average life expectancy of the population worldwide from 66.8 years in 2000 to 73.3 years in 2019 [1]. It is therefore of utmost importance to find strategies that can slow the aging process, while mitigating and/or preventing the diseases that are often associated with aging [2,3].

Aging is a complex and intraindividual process, usually defined as the time-dependent physiological loss of cellular integrity that leads to a deterioration of the body’s physiological functions [3]. Thus, for several years, many proposals have tried to understand and explain the aging process [4]. Two distinct theories have emerged, but together they establish a better understanding of the mechanisms involved in this phenomenon. The first theory suggests that there is an intrinsic and programmed deterioration of cellular functions [5], while the second theory pins down the existence of an accumulation of cellular damage, culminating in aging [6]. Under this rationale, López-Otín et al. [3] described this complex phenomenon under nine cellular and molecular markers of aging, including: (1) genomic instability; (2) telomere attrition; (3) epigenetic alterations; (4) loss of proteostasis; (5) dysregulated nutrient sensing; (6) mitochondrial dysfunction; (7) cellular senescence; (8) stem cell exhaustion and (9) altered intercellular communication. These markers are important throughout physiological aging, with the notion that, when they are intensified, there is an acceleration of the aging process [7].

The attenuation of the deleterious effect of aging can be accomplished through physical exercise (essentially resistance training and aerobic training). Garatachea et al. [8] summarized the multisystemic anti-aging effects associated with physical exercise, defining them under a multidimensional beneficial system: (1) increased neurogenesis and attenuation of neurodegenerative processes and cognitive changes; (2) decreased blood pressure levels and improved cardiovascular functions, such as increased cardiac output, blood volume, regional blood flow, improved endothelial and autonomic function; (3) improved respiratory function, with increased lung ventilation; (4) increased body metabolism, exceeding the basal metabolic rate, increasing protein synthesis and lipid oxidation; and (5) increased muscle synthesis, enabling improved strength, motor control, balance and joint mobility, as well as reduced body fat percentage with increased muscle mass and bone density. More recently, Rebelo-Marques et al. [7], based on the proposal by López-Otín et al. [3], placed physical exercise as one of the main strategies proven to have beneficial effects on each of the nine cellular and molecular markers of aging. Both these reviews [7,8] summarize the main benefits of physical exercise as an effective strategy that is practically free of serious adverse effects and capable of improving health and reducing age-related comorbidities (Figure 1). Despite the outstanding benefits of physical exercise and its few contraindications and risk of adverse effects, not all individuals can engage a regular and purposeful practice of exercise, mainly because of underlying pathologies or disabilities that make this practice unwise. It is thus imperative that alternative strategies be found, such as drugs and supplements, which can become an alternative (or complementary) option in these cases of impossibility of exercise practice. These pharmacological/supplementary alternatives do not aim to discourage the practice of exercise, but rather can be used as complementary entities to exercise and other existing therapies.

The development of pharmacological and nutritional strategies currently approved by the Food and Drug Administration (FDA) that prove to be effective in delaying this cellular degradation of particular aging-related diseases is an increasingly desirable target. Apparently, through strategies that allow delaying the aging process per se, it would be possible to concomitantly prevent the emergence of the great majority of age-related diseases, instead of fighting them individually. This framework is the central paradigm presented in this review. According to the current scientific evidence, we seek to review and analyze which individual therapeutic strategies have the ability to mimic or potentiate the beneficial effects of physical exercise in the aging process.

## 2. Drugs and Supplements

Several genetic, dietary and pharmacological interventions have been pinned down as useful tools within strategies that are capable of increasing longevity in the so-called “short-lived model organisms”, such as yeast, larvae, flies and mice [9].

A medicinal product—herein called a drug—is any substance or combination of substances presented as having properties for treating or preventing disease in human beings, or its symptoms, or that may be used in or administered to human beings with a view to making a medical diagnosis or, by exerting a pharmacological, immunological or metabolic action, to restoring, correcting or modifying physiological functions [10]. The value of each drug rests on four pillars: (1) therapeutic value; (2) preventive value; (3) health gains it brings; and (4) reduction in the costs of disease. It is based on this framework that researchers strive to find molecular and cellular targets in the field of anti-aging. Nevertheless, biological aging is closely related to changes in sensitivity to various drugs, changes in physiological energy reserves and response to therapies, proving to be a challenge in the use of these therapeutic tools [9]. 

A supplement is defined as a food, food component, nutrient or non-food compound that is purposely ingested in addition to the usual diet, with the goal of achieving a specific health and/or physical performance benefit [11]. 

Although science and technology in this field have evolved markedly in recent decades, the emergence of some drugs or supplements is still poorly studied, with scientific evidence on their safety and efficacy still very scattered and unsystematized. Aiming to fill some of the existing gaps and to help fill this gap in the literature, we herein summarize the effects of a set of drugs and supplements with the aim to delay the aging process (Figure 2). However, it is very important to note that some pathways and the molecular mechanisms largely remain to be clarified. A noteworthy example is the paradox of the circulating IGF-1 and the associated PI3K/AKT/mTOR signaling that affect cell proliferation, which is essential to antagonize the age-related loss of skeletal muscle [12].

## 3. Nutrient Sensing Regulators

The nutrient sensing regulators or neuroendocrine response modulators are composed of both substances of biological and synthetic origin. Rapamycin and its analogues—inhibitors of glucose metabolism and inhibitors of the GH/IGF-1 axis—are the main therapies addressed within this group. What fundamentally links this heterogeneous group of elements is related to the regulation of the nutrient sensing, leading the group with the greatest relevance and greatest depth of scientific knowledge on the subject of geroprotection. 

### 3.1. Rapamycin and Rapamycin Analogs (Rapalogs)

Rapamycin—also known as sirolimus—is a macrolide produced by a bacterium, *Streptomyces hygroscopicus*. Investigations in the late 1980s sought to identify the mechanism or mechanisms by which rapamycin blocks the growth of eukaryotic cells. Heitman et al. [13] has later discovered a protein—a target of rapamycin (TOR)—that would be responsible for the effects inhibiting cell growth. The TOR—a serine/threonine kinase—was considered to be the main compound in the response of eukaryotic cells to nutrients and growth factors, and this pathway became known as the TOR pathway [13]. Rapamycin was shown to form a complex with a protein, peptidyl-prolyl cis-trans isomerase FK506-binding protein 12 (FKBP12), and this was the complex responsible for TOR inhibition. Later, in 1994, three groups of authors identified mTOR, a catalytic subunit of two distinct complexes, mechanistic target of rapamycin complex 1 (mTORC1) and mechanistic target of rapamycin complex 2 (mTORC2), in mammalian studies (Figure 3).

The mTORC1 regulates a multitude of cellular processes, such as protein translation, autophagy, lysosomal biogenesis, lipid synthesis and cell signaling mediated by certain growth factors, through molecules belonging to the family of phosphoinositide 3-kinase kinases (PI3Ks) and Akt. The mTORC1 is further activated by nutrients (e.g., amino acids) and repressed by AMPK. On the other hand, mTORC2 is involved in processes such as cell cytoskeleton remodeling, cell survival, growth and proliferation, as well as ion transport. Inhibition of the mTOR subunit results in a variety of effects, such as induction of apoptosis and inhibition of cell progression through the cell cycle, cell growth, angiogenesis, endothelial cell proliferation and protein translation [14].

Rapamycin belongs to the first generation of mTOR inhibitors, which further comprises rapalogs, such as temsirolimus and everolimus, both of which also bind to FKBP12, showing improved pharmacokinetics compared to rapamycin. These three compounds thus have the ability to inhibit mTORC1. The second-generation compounds, such as NVP-BEZ235, PF-04691502, OSI-027, act by blocking adenosine triphosphate (ATP) binding to mTOR, also affecting mTORC2. There is also a third-generation of mTOR inhibitors, which target the various subunits that make up both mTORC1 and mTORC2 complexes, with high efficacy. An example of this category is the RapaLink-1 protein, but this is also the generation with the lowest robustness to date [15].

In 2009, Harrison et al. [16] found evidence that rapamycin increases longevity in mice, with increases of 9% in male mice and 14% in female mice. This was one of the major discoveries in the field of aging, as it was the first evidence that life could be significantly increased through the use of a pharmacological agent in mammals of both sexes [16,17]. Since 2009, there have been several studies on the anti-aging effects of rapamycin and its analogues, some of which are described in Table 1. Despite these findings, further studies are still needed, mainly human studies, in order to draw more definitive conclusions about its anti-aging effects, but also on safety of its short and long-term use.

### 3.2. Metformin

Metformin (dimethylbiguanide hydrochloride) is a derivative of the natural guanidines found in *Galega officialis*, a widely used drug that results in benefits in relation to glucose metabolism and diabetes-related complications. Increasing evidence in preclinical and human models suggests benefits of this substance in reducing the risk of age-related diseases such as neurodegenerative, cardiovascular and neoplastic pathology. These geroprotective properties, paired with its minor side effects, placed metformin under scrutiny, which has been gaining ground in the anti-aging industry, becoming one of the most promising strategies for treatment in humans. 

The exact mechanisms of action for an anti-aging effect are, however, not yet fully understood [15,33]. Although there is an improvement of molecular knowledge, metformin is a complex drug with multiple sites of action and multiple molecular mechanisms. It is known that is important for the activation of AMPK through the inhibition of complex I of the mitochondrial electron transport chain, although it can also activate AMPK through the lysosomal pathway [34,35]. Metformin can also inhibit mTORC1 independently of AMPK and have a significant impact on SIRT1 gene activation. The protein hypoacetylation capacity that metformin may possess is still under evaluation [36].

Returning to the cellular and molecular markers of aging proposed by López-Otín et al. [3], metformin represents a strategy that can primarily and most robustly cover four of the nine proposed markers, including (1) dysregulated nutrient sensing, (2) altered intercellular communication, (3) genomic instability and (4) loss of proteostasis. Secondarily, metformin shows benefits on the remaining five cellular markers of aging (Figure 4). The effects of metformin are mostly attributed to two components: a metabolic, which includes AMPK activation; and an oxidative, which includes inhibition of complex I of the mitochondrial electron transport chain. There are additional direct effects on molecules such as mTORC1, peroxisome proliferator-activated receptor gamma coactivator-1 alpha (PGC-1α), insulin/IGF-1, SIRT1, NF-kB and pro-inflammatory cytokines [33].

There are a few studies (Table 1) that highlight the benefits of metformin as mimicking or complementing the geroprotective effects of physical exercise [19,20]. Nevertheless, more robust studies are needed to reinforce the long-term safety of using this drug in humans, especially in non-diabetic individuals, so that there is evidence to expand its use as a sole anti-aging agent.

### 3.3. Glucose Metabolism Inhibitors

Several age-related metabolic disorders, such as type 2 diabetes, are mostly accompanied by disturbances in glucose homeostasis and cellular sensitivity to insulin [37]. Oxidative stress is considered one of the main complication induced by hyperglycemia, and is known to favor the formation of advanced glycosylation end products, markedly accelerating the aging process [38]. The non-degradation and subsequent accumulation of these glycosylation end products may be classified as reliable markers, not only of aging, but also of the so-called “metabolic memory”, referring to the persistence of adverse effects of hyperglycemia, even after normalization of blood glucose levels. In this sense, several studies seek to demonstrate that the inhibition of some of the enzymes of the glycolytic pathway may be the key to success in combating certain diseases directly associated with increased blood glucose levels and, consequently, aging [37].

The 2-deoxy-D-glucose (2-DG) is a glucose analog that naturally enters the course of glycolysis and has emerged as one of the most promising strategies in this area, despite its current lack of evidence robustness. This substance has a 2-hydroxyl group replaced by a hydrogen, and cannot undergo the glycolysis process completely. It is converted by hexokinase to 2-deoxyglucose-6-phosphate, which in turn will inhibit phosphoglucose isomerase and thus prevent the formation of glucose-6-phosphate. Thus, it is a promising strategy in areas such as antitumor therapy, since tumor cells exhibit high rates of glycolysis, using this metabolic pathway to generate ATP, their primary energy source [37]. 

The effects of chronic administration of a low dose of 2-DG in mice is presented in Table 1. Kumar et al. [21] proposed 2-DG to be an inducer of the formation of protective levels of ROS, enabling increased expression of autophagy-related genes (Beclina-1 and Atg-3), increased ferric reducing antioxidant potential (FRAP), mitochondrial complex I and IV activity, and molecules such as catalase (CAT) and superoxide dismutase (SOD), enhancing endogenous antioxidant capacity. Despite the advances, 2-DG does not have robust evidence that would allow its large-scale use, and further studies are needed to ensure strong levels of safety and efficacy.

### 3.4. GH/IGF-1 Axis Inhibitors

Among the signaling pathways associated with nutrient sensing, a major one is represented by insulin and IGF-1, playing a pivotal role in the regulation of aging and longevity. Intermittent fasting and protein and amino acid restriction may extend mammalian lifespan by modeling IGF-1 signaling [39]. Polymorphic variants of the IGF-1 receptor gene have been found to be linked to longevity in humans, in particular to exceptional longevity in centenarian individuals [40].

As is known, anterior pituitary hormone GH exerts an effect which binds to the GH hepatic receptors, inducing the production of IGF-1 via, the JAK-STAT pathway, primarily in the liver leading to cell proliferation and metabolic regulation. Thus, as reported by Russo et al. [12], on the one hand we have reduced insulin-IGF-1 receptor signaling that is beneficial for health; however, on the other hand, decreased IGF-1 with age is detrimental to skeletal muscle.

This paradox is still shrouded in controversy; nevertheless, it is suggested that healthy living can be maximized in young subjects by reducing IGF-1 levels, while the opposite occurs in old age [41].

Conover [42] proposed to act on pregnancy-associated plasma protein A (PAPP-A)—a zinc metalloproteinase—to model the IGF-1 availability. This substance is known for its ability to bind to glycosaminoglycans present on the cell surface, functioning as a growth-promoting protein as it increases the bioavailability of insulin-like growth factors in the vicinity of their receptors. Studies on mice knockout for PAPP-A showed a substantial increase in longevity, suggesting this important role of PAPP-A in aging and in the risk of age-related diseases [43,44]. However, despite being promising, this hypothesis is devoid of evidence that allows for complete safety in its use.

As an example, patients with acromegaly—an endocrine disease characterized by elevated GH/IGF-1 levels—have higher mortality rates. In this sense, medical therapy with somatostatin analogues or GH receptor antagonists, such as pegvisomant, emerged as promising strategies. However, both options require injectable administration and represent an expensive strategy without robust benefit in this context [37].

More recently, Slack et al. [45] identified an important role of the Ras-Extracellular signal-related kinase-E-twenty-six (Ras-Erk-ETS) pathway in reducing insulin/IGF-1 signaling. Specifically, early adulthood supplementation with an FDA-approved drug for the treatment of melanoma, trametinib, which is highly specific for the Ras-Erk-ETS pathway (Figure 5), has been shown to cause significant extension of longevity in *Drosophila* through its proven inhibition of cell growth and proliferation [46].

Despite the little evidence found, the group of GH/IGF-1 inhibitors are a promising target, but further studies are needed, especially in humans, to ensure efficacy and safety, and to allow conclusions to be drawn as to their real benefits in the field of geroprotection, especially in elderly frail persons or cachectic patients in whom quality of life is paramount. 

## 4. Autophagy Inducers

Autophagy inducers represent one of the most promising strategies to delay aging [47]. Autophagy phenomena ensure not only the homeostasis, but also the overall proteostasis of the cell. Through mechanisms that route toxic substances to a lysosomal degradation pathway, these processes detoxify the cell and recycle materials that have an accumulation throughout cellular aging [47,48]. As the main representative of this set of elements in this review, the spermidine is highlighted [22].

### Spermidine

Spermidine is a natural polyamine and the presence of this substance at the intracellular level is a result of its extracellular uptake, endogenous biosynthesis, catabolism and excretion. Some of the foods richest in spermidine are wheat germ, natto (fermented soybeans), soybeans, aged cheese, mushrooms, peas, nuts, apples, pears and broccoli. Spermidine biosynthesis is achieved through the formation of ornithine, a precursor of putrescine, which in turn is a precursor of spermidine. Their catabolism is dependent on the oxidative degradation of spermidine to spermine, and on the degradation and secretion of spermine, which requires their acetylation (dependent on acetyl-coenzyme A) by spermine-/spermidine-N1-acetyltransferase 1 (SSAT1) and subsequent oxidation. Polyamines also interconnect with substances that are relevant for amino acid metabolism. Spermidine synthesis requires the formation of decarboxylate S-adenosylmethionine (dcSAM), an important cofactor for protein methylation, including histone methylation, and is important for cellular epigenetic control. In addition, through the putrescine precursor, ornithine, polyamine biosynthesis affects the bioavailability of arginine, which is important to produce nitric oxide (NO), a molecule that mediates processes such as vasodilation, affects mitochondrial function and biogenesis, and has some immunomodulatory effects [48].

Interestingly, spermidine can improve mitochondrial function, possibly increasing the degradation of these organelles by mitophagy processes. These properties have led to several studies that sought to demonstrate the protective effects of spermidine treatment in various age-associated pathologies [22,49,50]. 

Spermidine administration has shown cardiac protective effects in mice, reducing ventricular hypertrophy and improving diastolic function, and, more importantly, intake correlates negatively with cardiovascular risk in humans. Noteworthily, spermidine reduces levels of gamma interferon, IL-1β, IL-6 and TNF-α in mice [22,51].

The basis of the molecular mechanisms that enable the effects of spermidine lie around its ability to induce autophagy (Figure 6). Spermidine has the ability to inhibit E1A binding protein P300 (EP300), an acetyl transferase, resulting in deacetylation of cytosolic proteins relevant to the autophagy process. Spermidine also possesses the ability to induce transcription of genes relevant to autophagy processes. This involves regulation of the transcription factor FOXO, as well as inhibition of histone acetyltransferases, with the whole process resulting in cellular reprogramming. Spermidine also has effects in suppressing tumorigenesis by inducing autophagy. In competent tumor cells, spermidine promotes ATP-dependent autophagic release, which in turn promotes immune surveillance. As for anti-inflammatory effects, these are explained by action on macrophages, promoting M2 polarization and suppression of NF-kB-dependent pro-inflammatory cytokines, which leads to decreased T-cell activation. At the same time, spermidine favors the formation of CD8+ T cells via induction of autophagy. Suppression of circulatory cytokines, such as TNF-α, also contributes to cardiovascular protection, possibly via a combined action with NO [48].

## 5. Senolytics

Fisetin, quercetin and navitoclax are the three main substances that represent the group of senolytic elements, all of which have the characteristic that, directly or indirectly, lead to the activation of mechanisms that eliminate the accumulation of senescent cells. By this mechanism, they enable the elimination of potential endogenous aggressors.

### 5.1. Fisetin

Fisetin (3,3′,4′,7′-tetrahydroxyflavone) is a flavonoid compound belonging to the polyphenol group and is found in various fruits and vegetables, such as strawberries, apples, persimmons, onions, cucumbers and grapes, in a concentration of 2–160 µg/g [52]. Fisetin acts, in part, through inhibition of members of the B-cell lymphoma 2 (Bcl-2) family—such as B-cell lymphoma-extra-large (Bcl-xL)—with anti-apoptotic properties, as well as through inhibition of other molecules, such as hypoxia inducible factor 1-α (HIF-1α) and other components of the senescent cell anti-apoptotic pathway (SCAP). Compared to other flavonoids, fisetin exhibits levels of senolytic efficacy that are often twice as high [53].

There are many flavonoids available that can yield an impact on the aging process. Yousefzadeh et al. [54] investigated which flavonoid demonstrated higher senolytic capacity, using murine fibroblasts with oxidative stress-induced and human fibroblasts genotoxic levels of senescence. Among the ten flavonoids evaluated, fisetin was the flavonoid with the most potent senolytic ability. The senotherapeutic capacity demonstrated by fisetin also suggested its feasibility for translation into human clinical studies. Within the same line, Zheng et al. [23] additionally demonstrated that fisetin was able to reduce microglial cell activation, PGE2 and NO production.

Although a potential translation is pinned down to human research, it is crucial to have protocols and clinical studies to clearly demonstrate the ability and safety of using fisetin for senolytic and senotherapeutic purposes.

### 5.2. Quercetin

Quercetin (3,3′,4′,5,7-pentahydroxyflavone) is a flavonoid found in some fruits and vegetables, such as capers, apples, berries, brassica vegetables, grapes, onions, shallots and tomatoes, as well as in nuts and green tea, in concentrations of 2–234 mg/100g. It is also found in some plants such as *Ginkgo biloba*, St. John′s wort and elderberry [55].

Existing evidence suggests that quercetin activates cell apoptosis through a mitochondrial pathway that involves the activation of caspase-3 and caspase-9 by the release of cytochrome c. It also has the capacity of modulating anti-apoptotic (Bcl-2 and Bcl-xL) and pro-apoptotic (Bcl-2 associated protein X (Bax)) proteins. The overexpression of Bax, associated with a release of cytochrome c and consequent translocation of factors to the cell nucleus, which induce apoptosis, seems to be one of the main effects of this substance [56]. Regarding an anti-inflammatory component, quercetin seems to inhibit inflammation mediated by TNF-α production. Quercetin prevents TNF-α from directly activating extracellular signal-related kinase (ERK), c-Jun NH2-terminal kinase (JNK) and NF-kB, which are potent inducers of pro-inflammatory gene expression. Quercetin also appears to indirectly inhibit inflammation by increasing peroxisome proliferator-activated receptor gamma (PPARγ) activity, thereby antagonizing the transcriptional activation of pro-inflammatory genes. Together, all these factors block the induction of inflammatory cascades mediated by TNF-α [55].

A human pilot study was conducted on 14 patients with idiopathic pulmonary fibrosis, an extremely devastating and progressively developing disease associated with accelerated aging of the alveolar epithelium. In this study, intermittent administration of quercetin (250 mg, 5 times/day) associated with dasatinib (100 mg/day), three times a week over three months, demonstrated an improvement in physical abilities, as assessed by increased distance walked (in meters), speed, and decreased rest time [26]. These data provide encouraging evidence for the development of randomized, controlled clinical trials in both idiopathic pulmonary fibrosis and other diseases associated with cellular senescence that are needed to provide robust evidence and proven safety to define the expanded use of quercetin in clinical practice.

### 5.3. Navitoclax

As mentioned above, the Bcl-2 family proteins function as regulators of programmed cell death, so the multiple interactions that may compromise or promote the activity of these same proteins will determine cell death or survival. Inhibitors of anti-apoptotic processes have been developed and studied extensively to achieve new therapies targeting age-related pathologies, such as certain types of cancer, diseases involving fibrosis and organ commitment processes, and some autoimmune diseases. Navitoclax (ABT-263) appears as one of these therapies, developed as a pro-apoptotic substance with affinity for members of the Bcl-2 family, which so far has only phase I and phase II studies. Navitoclax (ABT-263) is a Bcl-2 family inhibitor, developed based on a predecessor (ABT-737), a substance with a large molecular weight (800 g/mol), which contributed to poor cellular affinity, as well as to difficulty in metabolism. In addition to these characteristics that compromised its use, ABT-737 was not orally bioavailable. The modification to navitoclax allowed for better pharmacokinetics and pharmacodynamics, which enabled the potentiation of the effects of this substance [24].

Early studies with navitoclax showed promising results based on tumor suppression of a specific type of lung cancer, small cell lung carcinoma, as well as in acute lymphocytic leukemia [24] and non-Hodgkin’s lymphoma [25]. Navitoclax works as a BH3 mimetic, slinging to the BH3 (anti-apoptotic) domain of Bcl-2 proteins, which causes the release of Bcl-2-like protein 11 (BIM). This free state of BIM allows (for) its apoptotic action, centered on cytochrome c release, activation of caspases and ultimately cell apoptosis [57].

Elucidation of the potential of this substance, used alone or in combination, will allow a better and, above all, a rational and safe use. Its oral bioavailability may also potentiate a greater clinical use [58].

In addition to the studies presented in Table 1, further studies should be developed to explore and direct knowledge towards the specific mechanisms that enable the effects shown by navitoclax.

## 6. Telomerase Activators

Telomerase activation is essential for the maintenance of telomere length, to slow aging and reverse age-related tissue degeneration. Although telomerase plays an important role in slowing down the telomere attrition, the effect of physical activity and exercise are still not consistently demonstrated [59].

### TA-65

The TA-65 is a substance isolated from the *Astragalus membranaceus* root extract. The TA-65 has the capacity to increase the average telomere length and decrease the percentage of critically short telomeres and DNA damage in embryonic fibroblasts of haploinsufficient mice that harbor critically short telomeres and a single copy of the telomerase RNA component (TERC) gene [60]. A dietary supplementation with TA-65 in female mice also leads to improvements in several markers, such as glucose tolerance and osteoporosis, without significantly increasing overall cancer incidence [60].

A randomized, double-blind, placebo-controlled clinical study involving 117 relatively healthy cytomegalovirus-positive individuals (aged 53 to 87 years) showed that TA-65 can lengthen telomeres in a telomerase-dependent manner [27]. Another study, involving 7000 person-years over a 5-year period, found that TA-65 improves some metabolic, bone and cardiovascular health markers, without any adverse events, suggesting that it improves overall health by reducing mobility and mortality risks [61].

Regarding an immunoregulatory capacity, Liu et al. [62] pinpointed some positive remodeling effects, including a significant decline in the percentage of senescent cytotoxic T cells and natural killer (NK) cells, in cytomegalovirus-positive humans over the 1-year period. In human CD4 and CD8 T cells cultured from six healthy donors, TA-65 was found to increase telomerase activity by regulating the specific mitogen-activated protein kinase pathway MAPK.

Despite these studies, more attention and robust evidence are needed to evaluate the mechanisms underlying the action of TA-65, especially in the evaluation of its absorption, distribution, metabolism and excretion. However, based on the existing studies, TA-65 shows good potential for broad application in aging and related diseases.

## 7. Epigenetics

### Resveratrol

Sirtuins are NAD+-dependent protein deacetylases that play a crucial role in the epigenetic regulation of gene expression. In mammals, SIRT1, a homolog of Sir2 (present in yeast), has been found to act as an anti-aging molecule in the context of caloric restriction [63]. In addition to its gene silencing action, SIRT1 has shown an ability to regulate other important biological processes, including resistance to oxidative stress, cell survival, inflammation, mitochondrial biogenesis, vascular function and also apoptosis and autophagy [37]. Over the last few years, there has been extensive screening to find molecules or substances with sirtuin activation capacity. In this context, research on resveratrol has emerged.

Resveratrol (3,4′,5-trihydroxystilbene) belongs to the family of natural phytoalexins. It is present in various foods such as grapes, red wine, pistachio nuts, peanuts, blueberries and dark chocolate and is available in cis and trans isomeric forms. Only trans resveratrol is responsible for the effects on longevity and cardiovascular benefits [64]. Resveratrol is a remarkable activator of SIRT1. During aging, the gradual loss of SIRT1 expression is noticeable, especially in endothelial cells and cardiomyocytes, initiating vascular and cardiac aging. Being an activator of protein deacetylases, resveratrol will have the ability to indirectly deacetylate liver kinase B1 (LKB1), promoting the phosphorylation of AMPK. Resveratrol also has the ability to promote the deacetylation of transcription factors, such as NF-kB and FOXO, and also acts epigenetically as a histone deacetylase for transcription factors. This process of epigenetic regulation will be responsible for promoting, in the cells of the cardiovascular system, an internal environment free of oxidative stress and pro-inflammatory molecules (Figure 7). Within the scope of mimicking the effects of exercise, this substance also stands out in the increase of aerobic capacity, angiogenesis, mitochondrial biogenesis and activation of molecules such as PPARγ [64].

Pollack et al. [28] presented a randomized controlled, double-blind study in which elderly people with glucose intolerance were treated with resveratrol, 2–3 g/day, or placebo for a period of 6 weeks. The results showed that resveratrol had beneficial effects on vascular function, but not on glucose metabolism or insulin sensitivity (Table 1) [28]. Future studies in humans should address the appropriate dose range, as well as seek to circumvent the low bioavailability of resveratrol. 

## 8. Antioxidants

The oxidant aggressions to which we predispose our organism determine the central importance that is given to the group of antioxidants. Melatonin, of synthetic origin, along with curcumin, are the main elements that constitute the antioxidants group.

### 8.1. Melatonin

Melatonin (N-acetyl-5-methoxytryptamine) is an indoleamine, produced from tryptophan and released on a pulsatile pattern by the pineal gland, in close relationship with the circadian rhythm. It can be also synthesized by extra pineal tissues, such as the heart, liver, placenta, kidney, intestine and skin. Melatonin is an important regulator of physiological processes and is responsible for part of the body’s homeostatic balance. The level of melatonin throughout the day varies between 5–200 pg/mL, usually reaching its highest peak during the early morning hours. Among agents with antioxidant activity, melatonin stands out for its highly pleiotropic properties. This substance has antioxidant effects exerted by direct and indirect mechanisms, which make this molecule one of the most powerful strategies with robust capabilities, regarding its endogenous protective function against free radical formation [37,65].

Part of the actions of melatonin are believed to be mediated by the nuclear factor-erythroid 2-related factor 2-Antioxidant response elements (Nrf2-ARE) pathway. Generally, protection against oxidative stress is provided by direct and indirect antioxidant activities. Direct antioxidants are compounds that directly neutralize ROS. These compounds are themselves redox active and are consumed in the reaction with the ROS. Indirect antioxidants, on the other hand, activate natural redox systems, causing a transcriptional activation of a battery of cytoprotective proteins with catalytic action [66]. The nuclear factor-erythroid 2-related factor 2 (Nrf2) is a master regulator of indirect antioxidant activity. In response to oxidative stress, Nrf2 regulates protective genes by binding to antioxidant response elements (AREs) in the promoter regions of antioxidant genes. The SOD, CAT and glutathione peroxidase are among the genes activated by Nrf2. The Nrf2 interacts with other crucial regulators, such as NF-kB and p53, and is expressed in many tissues. The major organs of detoxification are the kidney and liver, which have the highest levels of this substance [37]. Studies in transgenic mice show that high levels of Nrf2 activity promote longevity, while low levels of activity are associated with a shortened lifespan, the justification being based on the ability of Nrf2 to positively influence pathways involved in cellular senescence processes [67]. 

Melatonin also exerts an anti-apoptotic effect, which depends on its ability to optimize mitochondrial function through antioxidant mechanisms. The anti-apoptotic effects of melatonin are associated with an increase in Bcl-2, as well as a decrease in Bax and caspase 3. However, an antiangiogenic and pro-apoptotic effect, through inhibition of vascular endothelial growth factor (VEGF), HIF-1α, Janus kinase/signal transducers and activators of transcription 3 (JAK/STAT3), was observed in liver carcinoma. Thus, melatonin is predicted to have a capacity to maintain pro- and anti-apoptotic balance, according to local needs, to maintain homeostasis [65].

The role of melatonin in the aging process is evident from the observation that both aging and related pathologies are closely linked to the loss of melatonin secretion and the decline in the circadian amplitude of melatonin. Recently, it has been assumed that the longevity-promoting effects of melatonin could be attributed to activation of SIRT1 [68].

In view of the above benefits, it is important to note that there is a need for solid data on the pharmacokinetic safety of exogenous administration of melatonin. Part of its beneficial effects may be attributed to the melatonin content of some foods that make up the Mediterranean diet, such as red wine, pistachios, olives and certain types of fish, and the doses required for the intended effects are still in question. Regarding supplemental administration, one of the key issues is therefore the dose needed for good efficacy of this substance, with minimal to no side effects [65]. 

Morganti et al. [29] demonstrated that the combination of melatonin with vitamin E and betaglucan was associated with reduced wrinkles, improved skin appearance and general well-being. There is a specific endogenous level below which melatonin is inefficient and, in such cases, exogenous supplementation may be necessary. Further studies are thus needed to strengthen the strategy of using melatonin as a pharmacomimetic for the benefits of exercise in physiological aging. 

### 8.2. Curcumin

Curcumin (1,7-bis(4-hydroxy-3-methoxyphenyl)-1,6-heptadin-3,5-dione), also called diferuloilmethane, is the main natural polyphenol found in turmeric. Turmeric is a spice that is receiving increasing interest both medically/scientifically and culinarily. It belongs to the ginger family and is classified as a *rhizomatous herbaceous perennial* plant. The interests in this spice have been known for thousands of years; however, the determination of the exact mechanisms by which it exerts its functions is relatively recent. Despite its reported beneficial effects, curcumin′s main problem is its low bioavailability, which seems to be mainly due to poor absorption, rapid metabolism and equally rapid elimination. Therefore, several agents have been tested to improve the bioavailability of this substance. One example is piperine, a known bioavailability enhancer, which is the main active component of black pepper and is associated with a 2000% increase in the bioavailability of curcumin [69].

Regarding mechanisms by which curcumin exerts its antioxidant functions, there is evidence that it can increase the activity of endogenous antioxidants, such as SOD and CAT, as well as serum concentrations of glutathione peroxidase and lipid peroxides [70,71]. It is also clear that curcumin can inhibit enzymes that normally generate ROS, such as lipoxygenase and cyclooxygenase, as well as xanthine hydroxygenase/oxidase [72]. The anti-inflammatory activity appears to be related to its ability to block activation of the transcription factor NF-kB [69]. 

A recent systematic review with meta-analysis [73] examined the effects of oral curcumin supplementation and found a significant effect on health-related quality of life. Cox et al. [30] added that curcumin can also improve mood and memory, as well as enabling the ability to learn in healthy individuals.

## 9. Adaptogens

### Rhodiola rosea

*Rhodiola rosea*, of the *Crassulaceae* family, is a plant traditionally used as an adaptogenic compound, and is also known as arctic root, rose root, rose orpin or golden root, and takes the form of a fleshy rhizome. It generally grows in limestone and granite soils at high altitudes (2000–5000 m). Adaptogenic effects have traditionally been referred to as inducing nonspecific immunity. In Asia, this plant has been incorporated into traditional Chinese medicine under the name *Hong Jing Tian*, where it is recommended to take 3–6 g of the root daily for increased vitality and longevity [74].

Within the compounds that constitute this plant, such as flavonoids, terpenoids, sterols and tannins, salidroside stands out as the main bioactive constituent. Salidroside (2-(4-hydroxyphenyl) ethyl-β-D-glycopyranoside) is a phenylpropanoid glycoside, which exists in greater amounts in the rhizome of *Rhodiola rosea* compared to the root, and in greater amounts in the male plant compared to the female. The main effects described for this substance are cardiovascular, central nervous system, anti-fatigue and anti-aging activity, as well as anti-inflammatory and antioxidant activity [75].

Wiegant et al. [31] sought to demonstrate, through studies with *Caenorhabditis elegans*, not only the increase in average life expectancy, but also how this phenomenon is justified. The authors showed that the mode of action of this adaptogen focuses on inducing the translocation of the transcription factor DAF-16 (an ortholog of the FOXO family, present in *C. elegans*), from the cytoplasm to the nucleus. This translocation suggests a potential for reprogramming transcriptional activities that favor the synthesis of proteins involved in stress resistance processes and also longevity. However, these results still lack translation to human organisms and there is a need for further studies to ensure which doses and timings of supplementation make *Rhodiola rosea* fit as one of the targets of anti-aging therapies.

## 10. Stimulants

### Caffeine

Caffeine (1,3,7-trimethylpurine-2,6-dione) is a natural alkaloid from the xanthine group, which can be found in beverages such as coffee, tea, energy drinks and various medications, as well as a myriad of dietary sources. This substance acts as a non-selective antagonist for the A1 and A2A adenosine receptors in the heart and brain and has antidepressant and diuretic effects [76].

Caffeine is known to affect cell growth, proliferation and energy metabolism by inhibiting the mTOR signaling pathway. TOR signaling, mentioned above in relation to rapamycin, is a fully known pathway in relation to maintaining homeostasis and cell growth, and caffeine exhibits a remarkably similar effect to rapamycin in inhibiting TORC1 and subsequently altering global gene expression patterns in yeast. Mutant cells lacking the genera encoding Tor1, Kog1 or Tco89, three specific nonessential components of TORC1, exhibit hypersensitivity to caffeine, suggesting that TORC1 is the specific target of caffeine [32,77,78]. However, this remains a controversial topic due to mixed results.

Lublin et al. [79] studied its effects on *C. elegans* adults at different concentrations to evaluate whether caffeine would be protective during aging. The authors demonstrated that caffeine (at a concentration of 0.1%) increased the maximum lifespan by 52% and significantly increased the mean lifespan in *C. elegans*. Their study also demonstrated that the protective effects of caffeine were completely blocked by inhibition of DAF-16, thus proving that the effects of caffeine are dependent on this FOXO orthologue present in *C. elegans*. Moreover, caffeine also reduced the acceleration of age-dependent death rate by 53%, and this effect was also blocked by DAF-16 inhibition.

Like rapamycin, caffeine also shows properties that allow it to target the mTOR pathway and its downstream cascade, proving to be an attractive target in the fight against aging [32,77].

Given the scarce scientific evidence on the use of caffeine as a geroprotective agent, especially in humans, it is essential to have further studies that investigate the potential effects of caffeine in delaying the aging process.

## 11. Current Gaps and Future Challenges

Through the results of this review, it is possible to state that a single cause could not adequately explain the appearance of codependent mechanisms that occur simultaneously with the physiological processes of aging. Understanding this network of causality allows the design of complementary strategies in order to minimize aging, which is, however, a difficult but at the same time very motivating goal. 

The research on strategies that symbolize gains identical to those that are given by physical exercise can provide important information for an expanded use on clinical practice. Within this line, using drugs or supplements to mimic or potentiate the well-known beneficial effects of physical exercise aging process becomes an arduous responsibility but often an almost unreachable mission [3,7].

It is correct to state that the prolonging effects of the anti-aging agents presented in this scope are complex and can be attributed to several cellular and extracellular signaling pathways. For this reason, considering the exceptional complexity of the mechanisms underlying these strategies, the identification and design of robust and safe anti-aging interventions seem to be tasks with many barriers but with positive long-term perspectives. 

On the other hand, the advances mentioned above are quite considerable and a relevant number of pharmacological substances with potential interaction in the cellular and molecular pathways underlying aging allow us to believe that the research, extremely important to outline and prospect the future, will not be in vain.

To aggregate, from an educational point of view, the promising strategies outlined in this review, we can fit them into three major groups. The first group consists of drugs and supplements that demonstrate anti-aging effects, but still without any evidence regarding their ability to extend lifespan. In the second group are those that suggest an extension of longevity, mainly because they can prevent or delay the progression of certain types of age-related diseases, such as cancer and cardiovascular disease, although they have not shown to slow the aging process itself. In the third group are the strategies that prolong cellular longevity, since they actually reverse the aging process itself, at least in particular situations [37]. Certainly, placing the strategies mentioned along our review in one of these groups would represent a simplified point of view since there are only a few studies available and, therefore, any new data that may appear could dictate the transition of a substance from one group to another. 

The implementation of these approaches in clinical practice will only be possible after a thorough examination and a more comprehensive discussion. To meet the needs caused by aging population, more emphasis needs to be placed on geroscience, which will require joint interdisciplinary efforts, active working researchers, physicians and other professionals seamlessly to make the transition from basic research to final clinical applications.

The major current limitation of evidence in this field is related to the scarcity of studies in humans, a fact that would be essential to reinforce the conclusions that were drawn from studies in animal species not only on efficacy data but also on safety data, namely the potential harmful effects of these drugs depending on dosage and other personal factors (such as age and frailty). In combination with this limitation, we highlight the significant heterogeneity on methodology of available studies—variable duration of treatment and follow-up, small sample sizes selected and studied—as well as the lack of a stratification of data that could significantly influence results and conclusions. 

## 12. Conclusions

Although animal models provide some important and preliminary data, more well-designed and robust studies are essential in order to determine with certainty which biological and/or synthetic therapies have the ability to mimic the anti-aging effects that physical exercise has been shown to harbor.

A healthy lifestyle, including a balanced diet, regular exercise and the disruption of certain intoxication pathways in the body are clearly first-line strategies to improve a healthy aging process. The use of potential and existing pharmacological substances can serve as an additional but not a substitute approach. 

The high expectations related to the development of these anti-aging interventions should, however, certainly be discussed and explored considering the economic, social and, crucially, ethical implications they entail. 

With an eye to the future, but with a firm grasp on the steps needed to achieve it, it is fair to say that current knowledge is just a tiny grain of sand. However, the quest to reach the whole desert of ignorance that lies behind the mechanisms of aging will be the cornerstone in the development of a different geroscience that we know today.

## Figures and Tables

**Figure 1 ijms-23-09957-f001:**
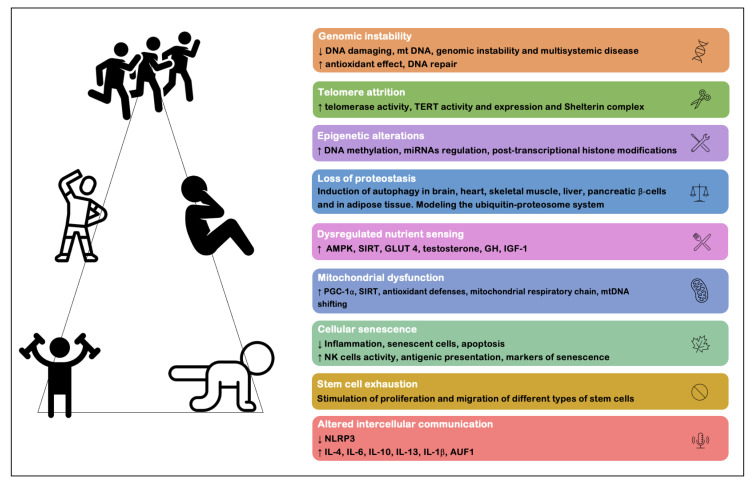
Benefits of exercise on nine cellular and molecular markers of aging (adapted from Rebelo-Marques et al. [7] and Garatachea et al. [8]). Abbreviations: AMPK, 5′adenosine monophosphate activated protein kinase; AUF1, AU binding factor 1; DNA, deoxyribonucleic acid; GH, growth hormone; GLUT4, glucose transporter type 4; IGF-1, insulin-like growth factor; IL-1β, interleukin 1β; IL-4, interleukin 4; IL-6, interleukin 6; IL-10, interleukin 10; IL-13, interleukin 13; mtDNA, mitochondrial DNA; NK, natural killer; NLRP3, NLR family pyrin containing 3; PGC-1α, peroxisome proliferator-activated receptor gamma-coupled receptor-1 alpha; SIRT, sirtuins; TERT, human telomerase reverse transcriptase.

**Figure 2 ijms-23-09957-f002:**
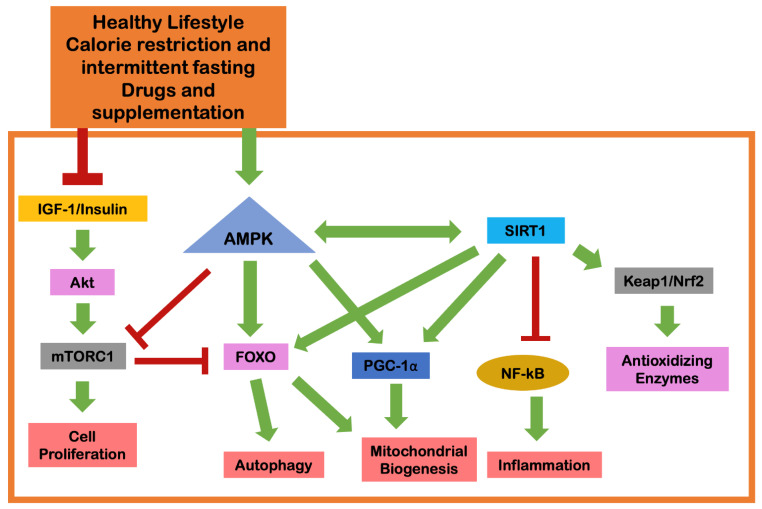
Molecular interactions that underlie the benefits of lifestyle, calorie restriction, intermittent fasting, drugs and supplements in geroprotection (adapted from Russo et al. [12]). The red bars in the figure indicate the inhibitory capacity of the molecular elements, while the green arrows refer to the capacity for stimulatory action. Abbreviations: Akt, protein kinase B; AMPK, 5′adenosine monophosphate-activated protein kinase; FOXO, forkhead box O; IGF-1, insulin-like growth factor 1; Keap1/Nrf2, Kelch-like ECH-associated protein 1/nuclear factor-erythroid 2-related factor 2; mTORC1, mechanistic target of rapamycin complex 1; NF-kB, nuclear factor kappa B; PGC-1α, peroxisome proliferator-activated receptor gamma coactivator-1 alpha; SIRT1, sirtuin 1.

**Figure 3 ijms-23-09957-f003:**
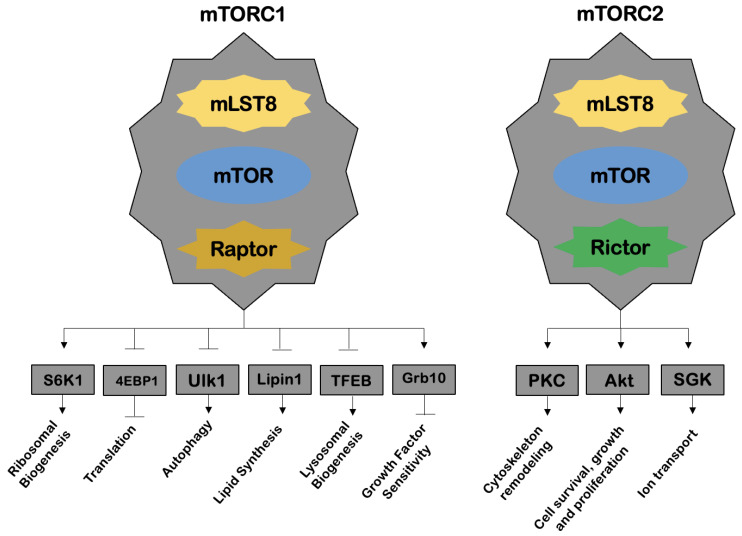
mTOR complex 1 (mTORC1) and 2 (mTORC2) components (adapted from Melick et al. [14]). The arrows represented in the figure refer to the stimulatory capacity of the complexes and, in contrast, the bars represent an inhibitory capacity. Abbreviations: mTORC1: mTOR, kinase (phosphorylation capacity); mLST8, positive regulator; Raptor, recognizes substrate. mTORC2: mTOR, kinase (phosphorylation capacity); mLST8, positive regulator; Rictor, recognizes substrate. 4EBP1, 4E-binding protein 1; Akt, protein kinase B; Grb10, growth factor receptor-bound protein 10; Lipin1, phosphatidate phosphatase-1; PKC, protein kinase C; S6K1, S6 kinase 1; SGK, serum glucocorticoid-induced protein kinase; TFEB, transcription factor EB; Ulk1, Unc-51 like autophagy activating kinase 1.

**Figure 4 ijms-23-09957-f004:**
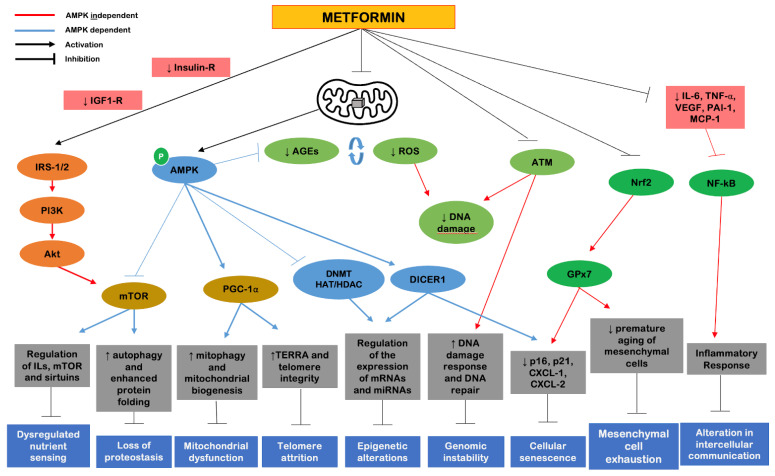
Mechanisms of action of metformin on cellular and molecular markers of aging (adapted from Kulkami et al. [33]). The red lines in the figure represent AMPK-independent mechanisms. The blue lines represent AMPK-dependent mechanisms. The inhibitory capacity of the molecular elements is given by the lines ending with a bar, while the arrows refer to the capacity for stimulatory action. Abbreviations: AGEs, advanced glycation end products; Akt, protein kinase B; AMPK, 5′adenosine monophosphate-activated protein kinase; ATM, ataxia-telangiectasia-mutated; CXCL-1, C-X-C motif chemokine ligand 1; CXCL-2—C-X-C motif chemokine ligand 2; DICER1, enzyme; DNA, deoxyribonucleic acid; DNMT, DNA methyltransferase; GPx7, glutathione peroxidase 7; HAT/HDAC, histone acetyltransferase/histone deacetylase; IL-6, interleukin 6; ILs, interleukins; IRS-1/2, insulin receptor substrate 1/2; MCP-1, monocyte chemoattractant protein-1; miRNAs, microRNAs; mRNAs, messenger RNAs; mTOR, mechanistic target of rapamycin; NF-kB, nuclear factor kappa B; Nrf2, nuclear-erythroid factor 2-related factor 2; p16, protein 16; p21, protein 21; PAI-1, plasminogen activator inhibitor-1; PGC-1α, peroxisome proliferator-activated receptor gamma-coupler-1 alpha; PI3K, phosphoinositide 3-kinase; ROS, reactive oxygen species; TERRA, telomeric repeat RNA; TNF-α, tumor necrosis factor alpha; VEGF, vascular endothelial growth factor.

**Figure 5 ijms-23-09957-f005:**
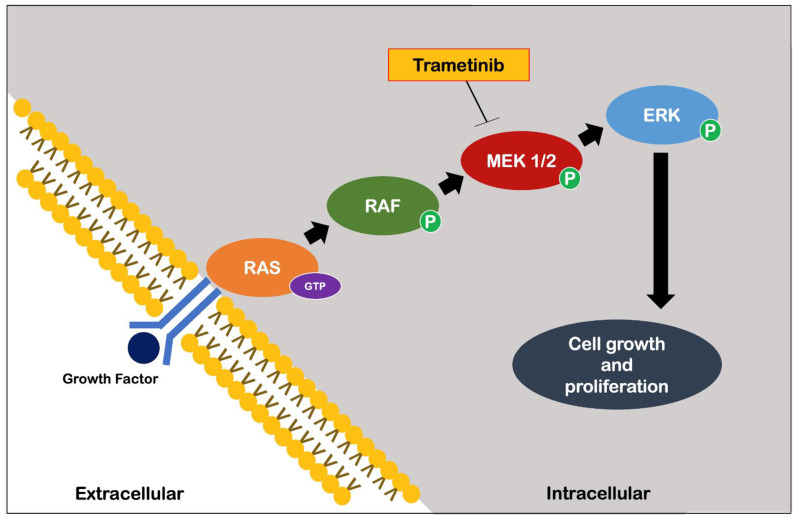
Trametinib action (adapted from Khunger et al. [46]). Arrows in black represent the stimulatory action of molecular elements. Abbreviations: ERK, extracellular signal-regulated kinase; GTP, guanosine triphosphate; MEK, mitogen-activated protein kinase; P, phosphorus; RAF, kinase; RAS, GTPase.

**Figure 6 ijms-23-09957-f006:**
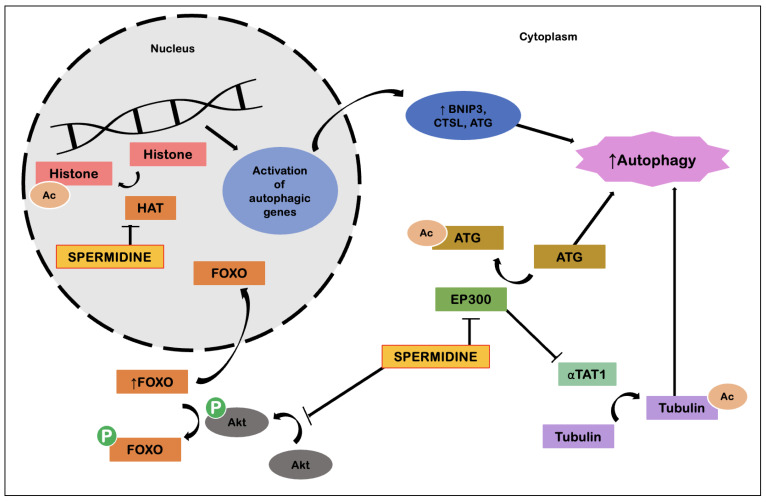
Spermidine-induced induction of autophagy (adapted from Madeo et al. [48]). Abbreviations: Ac, acetylation; Akt, protein kinase B; ATG, autophagy-related proteins; BNIP3, Bcl-2 interacting protein 3; CTSL, cathepsin L; EP300, E1A binding protein P300; FOXO, forkhead box O; HAT, histone acetyltransferase; P, phosphorus; αTAT1, alpha tubulin acetyltransferase 1.

**Figure 7 ijms-23-09957-f007:**
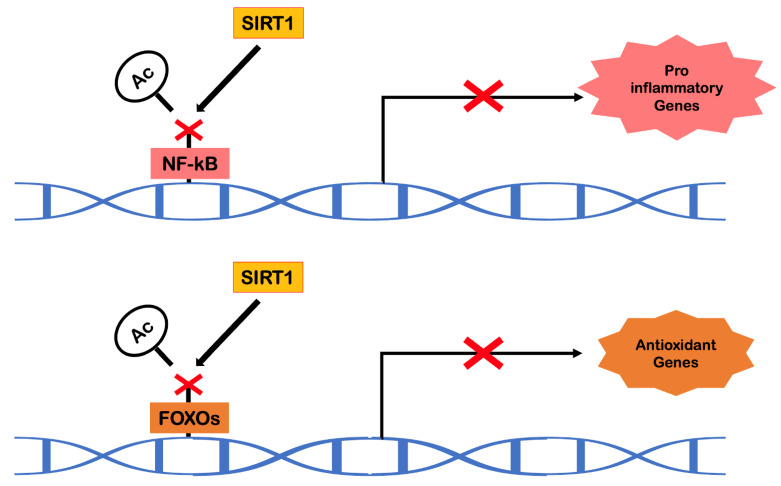
Transcriptional modulation of SIRT1 (adapted from Cheng et al. [64]). Abbreviations: Ac, acetylation; FOXO, forkhead box O; NF-kB, nuclear factor kappa B; SIRT1, sirtuin 1.

**Table 1 ijms-23-09957-t001:** Summary of the studies evaluated in terms of the anti-aging effects of the drugs and supplements discussed.

Groups	Substance	Original Source	Main Target	Model	Dose	Effect	*p* Value	Reference
**Nutrient sensing regulators**	Rapamycin	*Streptomyces* *hygroscopicus*	mTOR	Mouse	14 ppm	↑ longevity, 9% M – 14% F	<0.0001	[16]
				Human	0.2 µg/mL	↓ cellular senescence	<0.05	[18]
	Metformin	French lilac (*Galega officinalis*)	AMPK	Human	750 mg/day; weekly increase up to 1500–2250 mg/day	↑ AMPK phosphorylation; ↓ tumor cell proliferation	<0.05	[19]
				Human	850 mg	↓ coronary atherosclerotic risk	=0.02	[20]
	2-deoxy-D-glicose	Synthetic	Glycolysis	Mouse	25 mg/kg	↑ protective levels of ROS, FRAP, CAT, SOD + expression of autophagy related genes	<0.05	[21]
**Autophagy Inducers**	Spermidine	Wheat germ, natto (fermented soy), soy, aged cheese, mushrooms, peas, nuts	EP300	Human	Diet with high levels of spermidine-rich foods	↓ 40% heart failure risk + ↓ NT-pro BNP levels	=0.001	[22]
**Senolytics**	Fisetin	Strawberries, apples, persimmons, onions, cucumbers, grapes	SCAP	Mouse	0.5–2 µg/mL	↓ mycroglial cell activation, PGE_2_ and NO production	<0.05	[23]
	Navitoclax	Synthetic	Bcl-2 Family	Human	1 nmol/L	Inhibition of Bcl-xL activity, with translocation of Bax and cytochrome release, with cell apoptosis	<0.05	[24]
				Mouse	100 mg/kg/day, associated with bendamustine	Cell growth inhibition and delay in tumor growth	<0.05	[25]
	Quercetin	Capers, apples, berries, brassica vegetables, grapes, onions, shallots, tomatoes, walnuts, green tea	SCAP	Human	Quercetin: 1250 mg/day + Dasatinib: 100 mg/day	↑ physical abilities, as measured by increased distance traveled, speed, and decreased rest time	<0.05	[26]
**Telomerase Activators**	TA-65	*Astragalus membranaceus*	Telomerase	Human	250 U	↑ telomere length	=0.005	[27]
**Epigenetics**	Resveratrol	Grape, red wine, pistachio, peanut, blueberry and dark chocolate	SIRT1	Human	2–3 g/day	↑ vascular function	=0.002	[28]
**Antioxidants**	Melatonin	Pineal Gland	Nrf2-ARE	Human	1.6 mg de melatonin +1.6 mg de Vit. E + 1.6 mg de betaglucan	↓ wrinkles, improved skin appearance	<0.05	[29]
	Curcumin	*Curcuma longa* L.	Endogenous antioxidant system and NF-kB	Human	80 mg	↑ working memory performance, lower scores on fatigue, tension, anger, confusion, and mood disturbance	<0.05	[30]
**Adaptogens**	*Rhodiola Rosea*	-	DAF-16/FOXO	Nematode	10–25 µg/mL	↑ longevity; 10–20%, *C. elegans*	<0.001	[31]
**Stimulants**	Caffeine	Coffee, green tea	mTOR	Nematode	0.1% concentration	↑ longevity	<0.01	[32]

Abbreviations: %, percent; µg/mL, micrograms per milliliter; AMPK, 5’adenosine monophosphate-activated protein kinase; Bax, Bcl-2 associated protein X; Bcl-2, B-cell lymphoma 2; Bcl-xL, B-cell lymphoma-extra-large; CAT, catalase; DAF-16/FOXO, DAF-16/Forkhead box O; EP300, E1A binding protein P300; F, female; FRAP, ferric reducing antioxidant potential; g/day, grams a day; M, male; mg, milligrams; mg/day, milligrams a day; mg/kg, milligrams per kilogram; mg/kg/day, milligrams kilogram a day; mTOR, mechanistic target of rapamycin; NF-kB, nuclear factor kappa B; nmol/L, nanomoles por liter; NO, nitric oxide; Nrf2/ARE, Nuclear factor-erythroid 2-related factor 2/antioxidant response elements; PGE2, prostaglandin E2; ppm, parts per million; ROS, reactive oxygen species; SCAP, senescent cell anti-apoptotic pathway; SIRT1, Sirtuin 1; SOD, superoxide dismutase; U, atomic mass unit.

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
