# Peer review of "Therapeutics That Can Potentially Replicate or Augment the Anti-Aging Effects of Physical Exercise"

_ijms, 2022, doi:10.3390/ijms23179957_

Round 1
Reviewer 1 Report
The authors purpose an interesting view of ageing, but:
In figure 2 Cell Proliferation is not a cause of senescence, but rather negative if a neoplasia is present, but cell regeneration is positive, especially in maintaining muscle mass
It should not confuse the insulin and IGF1 mechanism, being the first consequent to glycemic peak in the second, for example, to exercise.
So metformin is effective in regulating glycemia rather than the mTOR mechanism.
An inhibitor of GH/IGF1 is related to cancer, but in normal condition is a positive effect.
in fig 6 correct spanish word
in my opinion, Melatonin has too little research to be cited as antioxidant.
Caffeine as well being an adenosine agonist it seems strange that could inhibit mTOR, you should cite more manuscripts or erase it
In conclusion, you should better address some issues (mTOR and IGF1 in particular) It is completely different to manage the elderly and a tumor!
It should be considered the overall action of polyphenols (not only quercetin or resveratrol....), for example considering the regulation o miRNA synthesis (see for example: 10.3390/antiox10020328)
Author Response
Thank you for your time revising our review and your thoughtful comments.
Attached you can find all our comments and changes.

Reviewer 2 Report
Manuskrypt ID: 1823138 , title: Therapeutics that can potentially replicate or augment the ijms anti-aging effects of physical exercise he is well prepared. Only a small tweak needed
1. The Methods of the review is not described. A description of methods is needed. What tools were used for the articles in the manuscript, the identification of relevant studies, the inclusion / exclusion criteria of articles, the selection of studies for inclusion and the evaluation of the strengths and limitations of the articles used.
2. Please add more keywords to increase access for readers
Hopefully these comments can help you improve your manuscript
Author Response

(The authors gave the same response as above.)

Round 2
Reviewer 1 Report
The authors have made changes as required, so the manuscript can be published